# Mapping Urban Air Quality from Mobile Sensors Using Spatio-Temporal Geostatistics

**DOI:** 10.3390/s21144717

**Published:** 2021-07-09

**Authors:** Yacine Mohamed Idir, Olivier Orfila, Vincent Judalet, Benoit Sagot, Patrice Chatellier

**Affiliations:** 1COSYS-PICS-L, Gustave Eiffel University, IFSTTAR, F-78000 Versailles, France; olivier.orfila@univ-eiffel.fr; 2COSYS-LISIS, Gustave Eiffel University, IFSTTAR, F-77454 Marne-la-Vallée, France; patrice.chatellier@univ-eiffel.fr; 3ESTACA Engineering School, F-78066 Saint Quentin en Yvelines, France; Vincent.JUDALET@estaca.fr (V.J.); Benoit.SAGOT@estaca.fr (B.S.)

**Keywords:** spatio-temporal geostatistics, mobile sensors, air quality, ozone concentration

## Abstract

With the advancement of technology and the arrival of miniaturized environmental sensors that offer greater performance, the idea of building mobile network sensing for air quality has quickly emerged to increase our knowledge of air pollution in urban environments. However, with these new techniques, the difficulty of building mathematical models capable of aggregating all these data sources in order to provide precise mapping of air quality arises. In this context, we explore the spatio-temporal geostatistics methods as a solution for such a problem and evaluate three different methods: Simple Kriging (SK) in residuals, Ordinary Kriging (OK), and Kriging with External Drift (KED). On average, geostatistical models showed 26.57% improvement in the Root Mean Squared Error (RMSE) compared to the standard Inverse Distance Weighting (IDW) technique in interpolating scenarios (27.94% for KED, 26.05% for OK, and 25.71% for SK). The results showed less significant scores in extrapolating scenarios (a 12.22% decrease in the RMSE for geostatisical models compared to IDW). We conclude that univariable geostatistics is suitable for interpolating this type of data but is less appropriate for an extrapolation of non-sampled places since it does not create any information.

## 1. Introduction

Air pollution is one of the major concerns of the last century and has caused more than 7 million deaths per year [1]. The situation is more alarming in metropolitan areas where the air quality regularly exceeds the standards suggested by the World Health Organization [2]. This can be attributed to the scale of urbanization and population growth, as well as the resulting energy consumption [3]. Air quality monitoring is a crucial part in the process of reducing urban air pollution and its harmful effects on people’s health and the environment. Indeed, real-time information on air pollution in urban areas is of great importance for environmental and health protection agencies who must advise the general public as soon as possible. This information can also be used by companies to offer several services and solutions in order to reduce the impact of air pollution on health.

### 1.1. Classical Methods of Air Quality Monitoring

Currently, air quality monitoring is carried out using fixed air quality monitoring stations. These stations are managed by national environmental protection agencies. These reference stations provide a very precise measurement of air quality at the cost of limited spatial coverage. The stations can generate detailed time series data, but only at limited locations. This makes it difficult to compile reliable and representative information for a city or a region as a whole, and therefore a more macroscopic view of trends in pollution fields is provided. However, the air quality in a city varies greatly because the concentration of pollutants in a given place depends mainly on local emission sources and atmospheric flow conditions [4].

For example, after comparing surveillance data from two streets in Copenhagen (*Jagtvej* and *Bredgade*), Berkowicz et al. [5] argued that roadside readings were site dependent and not representative of a larger urban area. They demonstrated that the measured concentrations could be very different at these two sites. Another study [6] showed that the air quality measurements taken at the intersection of two central London streets were highly dependent on the local wind flow and the geometry of the streets and buildings surrounding the receiver.

The total number of fixed air quality monitoring stations in a city is limited due to practical constraints, such as the cost and size of equipment and the power supply. An increase in the number of fixed stations is often hard to achieve. Hence, it is necessary to use other measurement and modelling techniques to assess urban air quality at unsampled places. There exist five large families of models and methods for creating urban air pollution cartography:
**Land-Use Regression models** Land-Use Regression models (LUR) make the assumption that the air quality in a given place depends only on the local characteristics of the environment, such as the land use, weather-related variables, building density, and traffic density. These models link the measurement of air quality taken at the fixed station to the chosen predictive environmental variables.A LUR model developed by Kerckhoffs et al. [7], including small-scale traffic, large-scale address density, and urban green, explained 71% of the spatial variation for ozone concentrations. Meng et al. [8] and Chen et al. [9] successfully developed a LUR model for NO2 concentrations in China.LUR models provide good results for a rather low complexity. They also describe the effect of the environmental variables on the pollutant concentration but remain limited by the amount of data from other variables needed or obtained at a relatively expensive cost.**Deterministic interpolation methods** One of the most popular deterministic interpolation methods is Inverse Distance Weighting (IDW). The value at the unknown location is calculated as the weighted average of the measurements collected from the monitoring stations. This method assumes that the value is more influenced by the nearest measurements than the distant ones, and thus the closest locations obtain greater weights. As the distance increases, less weight is given to the measurement.Given the simplicity of this method, it is often used as a benchmark. Marshall et al. [10] used it to compare the urban variability of the NO and NO2 concentration to a LUR model and an Eulerian grid model in Vancouver, Canada. Wong et al. [11] compared different interpolation methods, including IDW to estimate the ozone concentration and Particulate Matter (PM) concentrations.The weakness of deterministic interpolation methods lies in their poor extrapolation accuracy. These methods are not considered as models, because they do not describe the data in addition to not giving uncertainty associated with the prediction.**Geostatistics** Geostatistics regroup stochastic kriging methods, the value at the unsampled location is evaluated by a weighted linear combination of measurements, and the weights are calculated from the variability of the data inferred from the actual structure of the data.Kim et al. [12] developed an Ordinary Kriging (OK) prediction model to predict long-term PM concentrations in seven major Korean cities. Whitworth et al. [13] modelled the ambient air levels of benzene in an urban environment. More sophisticated than IDW and regression modelling, geostatistics also provide the uncertainty associated with the prediction. However, these techniques suffer from a relatively high computational cost.**Dispersion models** Dispersion models replicate the formation of atmospheric pollutants through physical and chemical processes. They have been widely used in traffic-related pollution prediction and make use of the environmental variables, such as the ones used in LUR models.Hamer et al. [14] described the Eulerian urban dispersion model EPISODE and its application to the modelling of NO2 pollution concentration. Fallah et al. [15] improved the characterisation of near-road air pollution using a regional Gaussian dispersion model. Gibson et al. [16] used the AERMOD Gaussian plume air dispersion model to evaluate the PM, NOx, and SO2. However, these methods suffer from numerous shortcomings, such as the computational cost and the production of uniform and imprecise maps, related to the challenging task of modelling the small scale random variations.**Machine learning algorithms** A machine learning algorithm analyses the training data and produces an inferred function, which can be used to map new examples. Machine learning is very effective in situations where insights must be discovered from large sets of diverse and changing data. Numerous studies applied this method to predict air pollution levels: Singh et al. [17] identified pollution sources and predicted urban air quality using ensemble learning methods. Cabaneros et al. [18] provided a review of Artificial Neural Network (ANN) models for ambient air pollution prediction. Some machine learning algorithms were combined with fuzzy models in order to predict air pollution levels [19]. Machine learning algorithms are considered as black boxes with poor descriptive power and struggle to provide better results than the other models with limited data.

With recent technological advances, the proliferation of air quality low-cost sensors offers additional tools to refine the spatial-temporal characterization of air pollution levels [20]. Numerous instruments from business entities, non-profits, and startups have entered the market thus far [21]. The performance of these sensors can differ significantly between different models as well as between units of the same model, as indicated by field and laboratory evaluations [22].

Although having many advantages, the use of this new type of sensors to assess urban atmospheric pollution also presents inconveniences. Mainly, taken separately, the data from these sensors are often noisy and not very precise. Studies [23,24] analysed the performance of low-cost air quality sensors as well as their benefits and their viability for monitoring air pollution levels in urban areas. None of the sensors tested showed good correlation with reference data in low ambient concentrations (0 to 15 μg/m^3^ range). When deployed in large quantities and using the right calibration and prediction models, they are able to provide complex and complementary information to the fixed monitoring station.

### 1.2. Mobile Sensors

The use of a fleet of low-cost sensors onboard vehicles (cars, buses, trams, and so on) travelling in an urban area in order to have a better representation of pollutants is increasingly popular. As opposed to the traditional air quality monitoring stations, the use of a low-cost mobile sensor network that can dynamically travel through the environment will deliver data with unprecedented resolution [25,26]. Some notable examples of research projects using low-cost sensors for monitoring air pollution include: the “OpenSense” projects in Switzerland [27], “Array of Things” in Chicago, United States [28], the Imperial County Community Air Monitoring Network [29] in California, United States, “Gotcha” II in Shenzhen, China [30], and the “Air Map Korea Project” in major cities of South Korea.

In this context, a mobile sensor could be a good compromise between temporal resolution and spatial resolution, allowing high spatial cover over large areas without using a large number of fixed sensors. However, due to the reduced temporal resolution of any sampled location, it is challenging to generate pollution maps with high temporal resolution at daily or hourly time scales.

Air quality monitoring using mobile sensors is attracting an increasingly growing interest [31,32]. Several devices have been developed to monitor, in real-time, the spatial and temporal variability of air quality using different instruments, technologies, and platforms. Gozzi et al. [33] summarized the status of mobile monitoring of PM. Most of these studies used mobile monitoring to assess air pollution exposure or to study spatial and temporal characteristics. Only a few studies were interested in producing urban pollution maps using mobile monitoring at a fine spatial-temporal scale.

A range of methods exist to go beyond the spatial and temporal coverage of the mobile measurements and draw pollution maps. Studies naturally applied the same methods used for fixed stations to the new problem generated by the use of mobile sensors. Table 1 summarizes the main recent studies using mobile monitoring to map air pollution levels.

Land-Use Regression models have become the standard method. Hatzopoulou et al. [51] and Kerckhoffs et al. [52] have evaluated the robustness of LUR models developed from mobile air pollutant measurements and concluded that mobile monitoring provided robust LUR models for predicting ultrafine particles concentrations. This partially explains the popular use of these models in mobile monitoring. All the studies in Table 1 have proposed models that share the same weaknesses with the LUR models: they require (and are mainly based) on information provided by external variables.

These variables are introduced into the model to investigate the link with the pollutant level, and the predicted pollutant value at unsampled locations is, therefore, derived from the knowledge of these variables at those locations. In addition to being able to predict only at the locations sampled by these covariates, the difficulty of their acquisition as well as the additional computational cost represent real obstacles to the use of these methods. Moreover, they have the disadvantage of producing maps with relatively large spatial and temporal resolutions. The final resolution of the prediction highly depends on the resolution of the covariates.

The problem worsens when we are interested in real time prediction. Either these covariates are sometimes available only after a given period of time, which makes them unavailable for real time prediction, or we use the predictions of these variables, which can introduce a lot of uncertainties in the final result.

Geostatistics have the advantage of being able to incorporate covariates (Kriging with External Drift (KED), Cokriging) but can also do without it (Simple Kriging (SK)), and thus represent, with the deterministic methods, a way to produce maps without using other variables. This method has the advantage, compared to the deterministic interpolation methods, to give the uncertainty associated with the prediction. However, geostatistics make stronger assumptions about the data. This model family was selected to tackle the real-time prediction problem because of the previously introduced advantages.

Some studies used geostatistics as a way to map air pollution using low cost mobile sensors. Li et al. [39] and Guan et al. [48], on top of using several covariates in their geostatistical model, used a likelihood-based method making stricter assumptions about the underlying distribution of the data and increasing the computational resources, making it challenging to use in real-time applications. Gressent et al. [43] used, as opposed to the likelihood method, a variogram-based method. They chose a purely spatial model that did not take into account the temporal correlation of the data.

This paper aims to show the prediction efficiency of variogram-based spatio-temporal geostatistics in the mapping process of air quality using mobile sensors without the use of external variables other than pollution data for real-time prediction purpose.

## 2. Materials and Methods

### 2.1. Data

Considering the limited number of studies carried out on urban air pollution with mobile sensors, the number of public datasets is limited. In this paper, we used the data from the OpenSense project to answer the research question. The ozone concentration was selected as the first pollutant to be examined in this study, and the methodology remains the same for any other pollutant categories.

The OpenSense project [53], is a Swiss project aiming to integrate air quality measurements from heterogeneous mobile and crowd sensed data sources in order to understand the health impacts of air pollution exposure and to provide high-resolution urban air quality maps. This project deployed several mobile air quality sensors on the trams’ roofs in the Swiss city of Zurich and Lausanne’s buses, collecting the measurement of ozone concentrations and counting Ultra Fine Particles (UFP). More information about the data as well as the data collection methodology can be found in these studies [37,54]. Even if these data show drawbacks, especially the sampling only on static trajectories of the city, they remain, nonetheless, very valuable for the application and the evaluation of new approaches to model the spatio-temporal variability of pollution in the urban environment.

In this paper, our study was carried out using the measured ozone concentration provided by the mobile sensors deployed on the top of the Zurich trams. The trajectory of the trams can be seen on Figure 1. Since the objective is to predict the concentration on a very detailed temporal resolution, this paper restricted the data used for a single week (from 28 February to 5 March 2016) containing data from five sensors on lines number 4, 7, 8, 12, and 13, resulting in a dataset of 40,000 observations.

The Opensense data provide the ozone concentration in parts per billion (ppb) in a given volume (volume of gaseous pollutant per $10^{9}$ volumes of ambient air). In order to convert it to μg/$m^{3}$ to match the unit of the data from the fixed monitoring station, we applied the following formula: μg/$m^{3}$ = (ppb)·(12.187)·(M)/293 where M is the molecular weight of the ozone pollutant ($M(O_{3})=48$). An atmospheric pressure of 1 atmosphere and a temperature of 20 ${}^{\circ }$C is assumed.

Reference data for fixed stations was obtained from www.ostluft.ch, the official air quality monitoring network in eastern Switzerland, which manages several fixed stations in the country. The data used here is the ozone concentration, available as hourly averaged. Since it is needed at a high temporal resolution, a linear interpolation was performed. The hourly averages were interpolated at each timestep when a measurement from the mobile sensors was collected.

#### Calibration Process

The data provided by OpenSense were raw and not calibrated. A first analysis showed that the sensor measurements differed significantly from each other even when they were close to each other. To reduce the bias and errors, a linear transformation using the data from the fixed monitoring station considered as aa reference was applied. The calibration was carried out separately for each sensor in order to achieve the best possible performance for the various sensors without changing their respective correlation.

Let $X_{i}\left(x,t\right)$ be the raw data coming from sensor *i* sampled at place *x* and time *t*, F(t) be the data from the fixed monitoring station at time *t*, and $Z_{i}\left(x,t\right)$ be the calibrated data from sensor *i* sampled at place *x* and time *t*.

A linear calibration of the raw data, to correct possible bias, is described as follows:(1)$$Z_{i}\left(x,t\right)=a_{i}+b_{i}\cdot X_{i}\left(x,t\right)$$

In Equation (Equation 1), the only known term is $X_{i}\left(x,t\right)$. The estimation of $a_{i}$ (additive bias) and $b_{i}$ (multiplicative bias) is needed to get the calibrated data. The estimation of $a_{i}$ and $b_{i}$ involves F(t):(2)$$F\left(t\right)=a_{i}+b_{i}\cdot X_{i}\left(x,t\right)+\epsilon$$

The estimation of $a_{i}$ and $b_{i}$ from Equation (Equation 2) was made using ordinary least squares, that minimized ϵ. This calibration was done for each sensor individually, using all sensor *i* data from all days in the dataset and fixed station data. There were as many estimates of $a_{i}$ and $b_{i}$ as there are sensors.

### 2.2. Methodology

As stated in the introduction above, there is a need for a method to generate real-time air pollution maps. In this section, the methodology used to assess the efficiency of spatio-temporal geostatistics is presented, by comparing different geostatistics models and show the potential gain compared to a standard IDW method, which is the most common and known in practice. First, the research question is defined:

What are the best models of space-time geostatistics for predicting urban air pollution using mobile sensors and what are the benefits compared to a standard deterministic approach? The remainder of this section develops each step of the methodology.

#### 2.2.1. Model Selection

Three geostatistical approaches were applied. Apart from mobile sensors data, two of them used fixed station data to predict air quality. Each of these three methods make different assumptions, which will be discussed in detail in the theoretical Section 2.3:Simple kriging with a varying known mean: the time series of the fixed monitoring station was chosen to be the overall mean.Ordinary kriging with a constant piecewise mean, but unknown.Kriging with external drift: the data from the fixed monitoring station was used to estimate the underlying mean.

The originality of the proposed models lies in their capacity to rely on a variographic study to describe spatiotemporal variance.

#### 2.2.2. Variographic Study

In this paper, only the estimation of the variogram and not of the covariance function was performed, making less restrictive assumptions on the stationarity of the random field. In the calculation of the experimental variogram, Arnaud et al. [55] recommend taking into account distances up to the half of the maximum distance encountered between two points in the field. Beyond that, the number of pairs of points involved in the calculation of the variogram decreases and reduces its robustness.

Knowing that, the maximum distance between two points in this study was 12.8 KM; variograms were, thus, limited to 6 km. As for the temporal limit, knowing that months of data were available, restricting this study to half of this temporal distance was neither possible in practice nor advantageous. The retained limit was set manually by increasing the time limit step by step until a sill appeared in the variogram.

One week of data was used to estimate the empirical variogram, all the data from this week was used for parameter estimation, which includes the 04/03 (the day of prediction) and the following day (05/03).

To study a possible anisotropy in the data linked to external factors, two spatio-temporal empirical variograms in the two static directions (north–south and east–west) were performed. Finally, three variograms were computed, each one associated with a different selected model.

#### 2.2.3. Models Validation Process

In order to evaluate the different models, a four-fold cross validation procedure was made, and the averages of the performance indicators used were computed. By varying the size of the training data set, conclusions about the efficiency of the models in different conditions are presented. Only the data from 04/03 was used in this cross validation procedure for the prediction/interpolation purposes following the three scenarios described below. The day 04/03 was chosen for the prediction tests for two main reasons: it is the day with the largest number of observations, and it represents teh typical daily ozone variation with a peak around 2 pm.

The data from 04/03 was kept in the parameter estimation procedure because, in practice, we did have access to a part of the data that we could include in the estimation of the variograms. Moreover, knowing that this cross validation procedure used different percentages of data, estimating a spatio-temporal variogram at each of these steps would be expensive in calculation cost. Furthermore, this data will not change much in practice in the estimation of parameters as it represents only a part of the global data used for parameter estimation (less than 1/5).

Three different ways for the random selection of points were chosen:The first method consists of randomly choosing a proportion of points regardless of their location in space or when they were collected: this corresponds to the reconstruction of data between sampled places.The second, more realistic, method consists of choosing small paths of different lengths while keeping the same percentage of data in order to reproduce a real data collection from a mobile sensor: this corresponds to the extrapolation of the data to places close to the sampling places.The last method, uses only the data resulting from the trajectory of specific trams. This corresponds to extrapolation for places "far" from the sampling points, which will often be encountered in practice.

#### 2.2.4. Performance Indicators

The three approaches were compared to one deterministic interpolation technique, here considered as the reference (IDW), in the three scenarios. The evaluation of the result of each of them used the following three performance indicators:The Root Mean Squared Error (RMSE) was selected as the main performance indicator to measure the error as it is the most frequently used measure to assess the differences between the predicted values by a model or an estimator and the observed values. The three geostatistical models presented in this article were built to minimize this error.
$$\mathrm{RMSE}=\sqrt{\frac{\sum_{i=1}^{n}{\left(Z_{i}^{*}-Z_{i}\right)}^{2}}{n}}$$The bias performance indicator was chosen to control the unbiasedness of the estimators. The three geostatistics estimators are theoretically unbiased. This performance indicator is used to check that.
$$\mathrm{BIAS}=1/n\sum_{i=1}^{n}\left(Z_{i}^{*}-Z_{i}\right)$$The correlation performance indicator was selected to deal with the low-cost nature of the sensors. In case of bias, it is necessary to measure the correlation performance and compare it to the RMSE.
$$\mathrm{CORR}=\frac{\sum_{i=1}^{n}\left(Z_{i}^{*}-\bar{Z^{*}}\right)\left(Z_{i}-\bar{Z}\right)}{\sqrt{\sum_{i=1}^{n}{\left(Z_{i}^{*}-\bar{Z^{*}}\right)}^{2}\sum_{i=1}^{n}{\left(Z_{i}-\bar{Z}\right)}^{2}}}$$

### 2.3. Methods

There are two ways of incorporating time into spatial geostatistics. The first is in the form of cokriging, and the second, more natural, by considering time as a separate dimension, which will be the case in this study. What has been considered here as support, is a unique sample measured in a volume of air.

Given a support *D* in $R^{n}$ and a probability space (Ω,A,P), a random function is a function of two variables Z(x,w) such that, for each *x* in *D* the section Z(x,.) is a random variable on (Ω,A,P).

In this case, $D=R^{2}\times R_{+}$ where $R^{2}$ represents space and $R_{+}$ time, the random function is simply denoted by Z(x,t), and a realisation of this random function is represented by z(x,t) where $x\in R^{2}\;and\;t\in R_{+}$.

The methods presented in this section have been theoretically defined in previous works [56]. However, the adaptation of this work to our use case required dedicated efforts. In the next section, we introduce the necessary theoretical details to understand the models.

#### 2.3.1. Simple Kriging with a Varying Mean

The application of simple kriging requires two hypotheses: the second order stationary of the random field, and the knowledge of the mean over the whole domain *D*. Assuming that the fitted data collected by the mobile sensors comes from a stationary random field of order two is a strong hypothesis that is not realistic. In this model, the data given by the fixed monitoring station F(t) is supposed to be the overall mean. Subtracting the value of the fixed station from the fitted data provided by the mobile sensors is assumed to be stationary of order two with a zero mean.

The simple kriging estimator is:(3)$$Z^{*}\left(x,t\right)=\mu +\sum_{i=1}^{n}\lambda_{i}\left(Z\left(x_{i},t_{i}\right)-\mu \right)=\sum_{i=1}^{n}\lambda_{i}Z\left(x_{i},t_{i}\right)$$
where μ is the mean of the detrended random field and is equal to zero. To produce the best linear estimator, we must ensure that the estimation variance is minimal and that the estimator is unbiased.

The unbiased condition is automatically verified, and does not imply any additional constraint because:(4)$$E\left[Z^{*}\left(x,t\right)-Z\left(x,t\right)\right]=\sum_{i=1}^{n}\lambda_{i}EZ\left(x_{i},t_{i}\right)=0$$

This leads to the simple kriging equations:(5)$$\sum_{j=1}^{n}\lambda_{j}\gamma \left(x_{i}-x_{j},t_{i}-t_{j}\right)=\gamma \left(x_{i}-x,t_{i}-t\right)\;i=1,..,n$$

The resolution of Equation (Equation 5) gives the different lambda in the linear combination (Equation 3).

#### 2.3.2. Ordinary Kriging

The application of ordinary kriging makes less restrictive assumptions—namely a constant but unknown mean. The linear estimator of ordinary kriging is written this way:(6)$$Z^{*}\left(x,t\right)=\sum_{i=1}^{n}\lambda_{i}Z\left(x_{i},t_{i}\right)$$

To ensure the unbiased condition:(7)$$E\left[Z^{*}\left(x,t\right)\right]=E\left[\sum_{i=1}^{n}\lambda_{i}Z\left(x_{i},t_{i}\right)\right]=m\sum_{i=1}^{n}\lambda_{i}$$
(8)$$\sum_{i=1}^{n}\lambda_{i}=1$$

The objective is to minimize the error, characterized by its expected mean square $E{\left(Z^{*}-Z\right)}^{2}$ under the unbiased condition (Equation 8) using the Lagrangian multiplier μ. The weights that minimize the error are the solution of: (9)$$\begin{array}{c} \begin{array}{cc} \sum_{j=1}^{n}\lambda_{j}\gamma \left(x_{i}-x_{j},t_{i}-t_{j}\right)+\mu & =\gamma (x_{i}-x,t_{i}-t)\;i=1,..,n\\ \sum_{i=1}^{n}\lambda_{i} & =1 \end{array} \end{array}$$

The equation system (Equation 9) is called the ordinary kriging system, and solving it yields the weights $\lambda_{i}$ for the linear estimator (Equation 6).

#### 2.3.3. Kriging with External Drift

Kriging with external drift or regression kriging assume that Z(x,t) can be broken down into two parts, one deterministic μ(x,t) and the other stochastic Y(x,t):(10)Z(x,t)=μ(x,t)+Y(x,t)
with *Y* being stationary intrinsic with zero mean. $f_{0},f_{1},f_{L}$ are deterministic functions with f:D⟶R, and μ(x,t) is a linear combination of these functions evaluated at (x,t):(11)$$\mu \left(x,t\right)=\sum_{l=0}^{L}a_{l}f_{l}\left(x,t\right)$$
with $f_{0}\left(x,t\right)=1$
(12)$$Z\left(x_{i},t_{i}\right)=\mu \left(x_{i},t_{i}\right)+Y\left(x_{i},t_{i}\right)=\sum_{l=0}^{L}a_{L}f_{L}\left(x_{i},t_{i}\right)+Y\left(x_{i},t_{i}\right)$$

The different functions $f_{l}\left(x,t\right)$ represents the covariates “external drifts” used to estimate the underlying mean; in this study, only one function $f_{1}\left(x,t\right)=F\left(t\right)$, which stands for the fixed station data, was used.

The linear kriging with external drifts estimator is, therefore, written:(13)$$Z^{*}\left(x,t\right)=\sum_{i=1}^{N}w_{i}Z\left(x_{i},t_{i}\right)=\sum_{i=1}^{N}w_{i}\left(\sum_{l=0}^{1}a_{l}f_{l}\left(x_{i},t_{i}\right)+Y\left(x_{i},t_{i}\right)\right)$$

The unbiased condition is satisfied if and only if:(14)$$\sum_{i=1}^{n}w_{i}f_{l}\left(x_{i},t_{i}\right)=f_{l}\left(x,t\right)\;l=0,1$$

Coupled with the minimum variance condition, this gives the kriging system (Equation 15):(15)$$\begin{array}{c} \begin{array}{cc} \sum_{j=1}^{n}\lambda_{j}\gamma \left(x_{i}-x_{j},t_{i}-t_{j}\right)+\sum_{l=0}^{1}a_{l}f_{l}\left(x_{i},t_{i}\right) & =\gamma (x_{i}-x,t_{i}-t)\;i=1,\cdots ,n\\ \sum_{i=1}^{n}w_{i}f_{l}\left(x_{i},t_{i}\right) & =f_{l}\left(x,t\right)\;l=0,1 \end{array} \end{array}$$

#### 2.3.4. Spatio-Temporal Inverse Distance Weighting

Inverse Distance Weighting is a type of deterministic method that assigns values to non-sampled points using a linear combination of values from sampled points weighted by the inverse distance.

The general formula for the IDW is given by Equation (Equation 16): (16)$$Z^{*}\left(x,t\right)=\sum_{i=1}^{n}\lambda_{i}z\left(x_{i},t_{i}\right)$$
with:(17)$$\lambda_{i}=\frac{1/d_{i}^{p}}{\sum_{i=1}^{n}1/d_{i}^{p}}$$

$d_{i}$ represents the distance between $Z^{*}\left(x,t\right)$ and $z(x_{i},t_{i})$. The weights decrease as the distance increases, especially as the power value *p* rises. As with the previous methods, points in the neighbourhood have a heavier weight and have more influence on the prediction, thus, resulting in a local spatio-temporal interpolation. In this study, this definition of a spatio-temporal distance was chosen:(18)$$d_{i}=\sqrt{{\left(x_{i}-x\right)}^{2}+{\left(y_{i}-y\right)}^{2}+C\cdot {\left(t_{i}-t\right)}^{2}}$$

The parameter *p* was fixed at 2, while *C* was obtained by cross-validation.

Finally, while any covariance function can be written in the form of a variogram using γ(h)=C(0)−C(h), the opposite is not generally true. The passage from variogram to covariance is only possible under the assumptions of second order stationarity.

This paper only uses the variogram and not the covariance function, making less strict assumptions.

## 3. Results

In this chapter, different results from the application of the methodology on the dataset are presented. Starting with the variographic study, we show the two different directional variograms, as well as the experimental variograms and their respective theoretical variograms considered for the different models. Then, a prediction with the three models was carried out for the day of 04/03 from 5 a.m. to 10 p.m. The result of the cross validation procedure in each of the scenarios is shown for the three performance indicators, for the three spatio-temporal geostatistical models as well as the IDW method. Last, the prediction of ozone concentration as well as the associated uncertainty via the KED model is displayed, using all the data available for one day.

### 3.1. Variographic Study

#### 3.1.1. Anisotropy

An isotropic phenomenon is a process that does not depend on any particular direction. In spatial studies, this process is considered to evolve in the same way in all directions. On the opposite, an anisotropic phenomenon is a process that varies in a different way depending on the studied direction. The anisotropy can be detected on the experimental variograms by different ranges according to the directions. Generally, it is observed that the directions of the longest and the shortest spans are orthogonal.

We calculated two spatio-temporal directional variograms using the pair of points in the north–south axis and in the east–west axis. The directional variograms were calculated from the fitted data without subtracting the fixed monitoring station values. Figure 2a,b shows that there were no significant differences between the two variograms.

#### 3.1.2. Spatio-Temporal Variance

The study of the spatio-temporal variability of the data showed a clear difference between the spatial and temporal variability. The different variograms showed that, on average, there was a greater difference between two measurements sampled a few hours apart at the same place, than two measurements sampled at the same time anywhere in space (on the scale of a city), which justifies the traditional approach using the fixed stations for monitoring air quality. Mobile sensors, in addition to being able to capture temporal variance, can also capture spatial variance.

As we do not sacrifice spatial variance by using them, we can only improve the explained global variance. The three variograms (fitted data in Figure 3, residuals in Figure 4, and estimated residuals in Figure 5) show exactly the same purely spatial variability. This is because, for residual variograms, we subtracted only temporal component provided by the fixed monitoring station, leaving the spatial variability unchanged.

The three computed empirical variograms show small nugget effects; however, there is no data at the same time and at the same place simultaneously as none of the trams meet. Moreover, the proximate collected data points necessarily come from the same sensor, and these measurements are not independent conditionally to the ozone concentration. This is why these variograms show small variability in the origin, which does not necessarily reflect the real variability of the studied phenomenon.

#### 3.1.3. Modelling

A metric theoretical spatio-temporal variogram assumes identical spatial and temporal covariance functions taking into account the spatio-temporal anisotropy:$$\gamma \left(h,u\right)=\gamma_{joint}\left(\sqrt{h^{2}+{\left(K.u\right)}^{2}}\right)$$
where $\gamma_{joint}$ is any known variogram that may include a nugget effect, and *K* is a spatio-temporal anisotropy parameter defined as the number of space units equivalent to one time unit. The estimation of *K* was done at the same time as all the other parameters of the theoretical model (i.e., the sill, nugget, and range) by minimizing the average of the squared deviations between the sample and the fitted variogram surface [57]. The used optimization algorithm is L-BFGS-B, which is the bound-constrained variant of the limited–memory Broyden–Fletcher–Goldfarb–Shanno optimisation algorithm. The different joint models and their respective parameters can be found in Table 2 for the three methods.

As expected, the variogram model associated with the ordinary kriging showed the highest range and sill, as opposed to the two other models, where the data from the fixed station partially explained the variance, resulting in a lower range and sill.

### 3.2. Spatio-Temporal Signals

Figure 6, Figure 7 and Figure 8 show the prediction for the different tram lines trajectories. In the first, second, and third scenarios. Only four sensors were functional that day: the sensors on the lines 4, 7, 8, and 13.

The first thing to notice is the similarity of the predictions of the three methods. This is explained by the same spatial variability common to the three variograms. Moreover, this spatial variability is smaller than the temporal one, and thus the three estimators mainly used the spatially close data. As the spatial variability did not change from one model to another, we found fairly similar predictions. The three estimators did not interpolate the data at the sampled locations; they are, therefore, not exact estimators due to the nugget effect, which represents measurement errors. The estimators, therefore, tended to filter the measurement errors.

In the third scenario (Figure 8), in the absence of data coming from the predicted tram line, the estimators tended to imitate the values sampled in the nearest tram lines. Thus, the prediction on line 2 was similar to the values sampled on line 17, and vice versa.

The inadequacy of predictions at a given location came from the lack of nearby data at that location, and this was more visible in scenario 3. The result was even worse at the end of the day. Indeed, in the absence of close data from the same tram, the predictions will be more influenced by the measurements taken at the same time by the other trams; however, we noticed a clear difference in the measurements taken at the end of the day.

### 3.3. Performance Indicators Results

The first thing to notice in the RMSE (Figure 9) is that the three probabilistic methods performed significantly better than the deterministic interpolation in each scenario. As expected, in the first scenario, the more data that were used, the less errors were made. This was not true in the third scenario where we noticed that the error reached a minimum. No matter how much data were used, the RMSE did not fall below 6 μg/$m^{3}$. The KED estimator showed the least errors in the case of data reconstruction at places close to the sampled data (scenario 1) followed by SK and OK. We concluded that the contribution of the fixed station data in such an environment was useful and that the KED optimized its use. In the two others scenarios, the use of ordinary kriging appeared to be more appropriate.

The four methods biases tended towards zero in the first scenario, and, in the third scenario, the prediction seemed systematically biased (Figure 10). Although the stochastic methods systematically outperformed the IDW method. This was not the case in the second scenario. The correlation Figure 11 consolidates the idea that KED seemed the best suited in the first scenario, where OK showed better correlation results in the second and third scenarios.

To summarize, in the first scenario, the performance indicators were smooth, and the more data we used, the better the predictions. This was not true in the third scenario, where we reached a sill regardless of the number of data points used. As for the second scenario, it was a mix of both.

We concluded that kriging using the data from the fixed measurement station as an external variable was the most suitable in the case of data interpolation. When we want to extrapolate far from the sampling places, ordinary kriging appeared to be the best solution. As expected, and as the majority of data reconstruction methods, geostatistics performed better in the case of interpolation versus extrapolation, regardless of the considered performance criterion.

### 3.4. Resulting Maps

To answer the objective of creating pollution maps, The KED algorithm was applied using every data point available from the mobile sensors, as well as the fixed monitoring station during one day. Figure 12 shows an example of 17 h of the resulting maps for 4 March 2016. Figure 12 display only the resulting ozone concentration from 5 a.m. to 10 p.m., when all four mobile sensors were active. The method succeeded in identifying areas with high ozone pollution in the city of Zurich, considering that only four mobile sensors were used. One of the important points that can be observed in Figure 12 is that the typical mid day spike of ozone concentration was clearly visible, followed by mostly very low concentrations during the evening and night.

The concentrations begin to increase throughout the city at around 6 a.m. (depicted by a brief peak observed on lines 13 and 7, as shown by Figure 6, Figure 7 and Figure 8). The concentrations reached a maximum at around 12 a.m./1 p.m., at this point, the resulting maps indicate concentrations exceeding 60 μg/$m^{3}$ along the north-west side of the city. Finally, the overall ozone concentration decreased again throughout the evening, and, around 7 p.m., reached approximately the same levels as during the previous night of around 20 μg/$m^{3}$ in most areas of the city.

As stated above, one of the advantages of geostatistical models is to provide prediction uncertainty, and Figure 13 shows the variance associated with the KED estimator. The kriging variance is not related to the data values, but only to the data placement; this is why there is no correlation between Figure 12 and Figure 13. The further away from the location of the collected data, the greater the variance, and vice versa. The relationship between the variance and the distance from the data was directly impacted by the spatial-temporal variogram.

As expected, the variance was minimal in the centre of the city where there was the most data collected. The locations of the four sensors can be easily seen at certain moments of the day (11 a.m. or 5 p.m.).

The maximum variance can be observed at the edge of the maps shown at 5 a.m. and 22 p.m. These two maps have the singularity of having data collected only on one side of the time, resulting in a great temporal distance (beginning and end of the day) on top of a great spatial distance (edge of the map) to the collected data, which, as said above, implied a greater uncertainty.

## 4. Discussion

In this paper, several findings need to be highlighted: Spatio-temporal geostatistics offers tools to deal with the problem of using mobile monitoring sensors. While other studies relied on several covariates to predict air quality, this approach can be used to create real-time air pollution maps. The advantage of geostatistics is that we are not restricted to a given temporal or spatial resolution. Therefore, we can predict at any distance step and any time step. It would also be possible to predict at greater scales, such as road sections or longer time periods using block kriging.

Despite the subtraction of the data coming from the fixed stations, there still exists a large spatio-temporal variability, which could be easily captured by mobile sensors as it can be seen in the results of this paper.

However, several limitations in this study must be detailed: The trams did not go through all types of streets and, therefore, only measured a specific type of urban pollution. Furthermore, the methodology described above was not used to identify the best model to estimate the ozone concentration but rather the concentration measured by sensors similar to those used in this study. In this dataset, we do not have access to the real value of the ozone concentration from reference sensors, and it is, therefore, impossible to carry out a cross validation for this purpose. Moreover, the data from the mobile sensors were considered independent conditionally on the ozone concentration, and this study did not take into account the autocorrelation of data from the same tram.

Ordinary kriging does not use the fixed station data in its prediction. Therefore, the geostatistical approach can be evaluated in the absence of other data except the ones collected by the mobile sensors. The assumption has been made that the mean is constant, but unknown, or at least locally constant, being equal to the average of a limited number of datapoints in the neighbourhood of the target point to predict. Thus, this approach is not completely independent from the fixed station data: actually, in the process of sensor calibration using an additive bias (Equation (Equation 1)), the empirical mean of each sensor is imposed to be equal to the mean of the fixed station. Knowing that the ordinary kriging assumes that the average of the field is constant and, therefore, tends towards the mean of the measurements coming from the mobile sensors, finally, the predictions from the ordinary kriging also tend towards the mean of the fixed station.

In this study, no model was capable of predicting a value that lay outside of the range of data points on which it was based. Since these interpolations are carried out on subsets of control data, the max and min values in those subsets will be the upper and lower limits of what the methods can predict.

As no relationship between the spatial coordinates and the variable of interest (ozone concentration) was found in this study, universal kriging could not be used. The absence of auxiliary variables makes the prediction outside the collection areas collection extremely hazardous. As geostatistics do not create information, one must rely on dependencies with other variables to predict pollutant concentrations outside the sampling area.

## 5. Conclusions and Perspectives

Air pollution maps with high spatio-temporal precision is of paramount importance and remains an unsolved problem. The use of a mobile sensors fleet, by increasing the spatial coverage, offers a solution to this problem. The use of these devices requires new models to manage these data and produce air quality maps. In this paper, we proposed the study of three spatio-temporal geostatistics methods, and, by comparing them to a deterministic interpolation, we concluded that the probabilistic methods systematically outperformed the deterministic method. The use of univariable geostatistics provided conclusive results and is more suitable for interpolation at places close to the sampling site.

For the extrapolation, it will be necessary to use auxiliary variables in the form of cokriging or regression-kriging. Despite a higher complexity, the anisotropic models could improve the quality of the prediction. In this paper, we only tested a fixed spatial anisotropy in time, another idea would be to search for a possible variation of anisotropy, related for example, to the wind speed and direction. Even if univariate geostatistics have its own benefits, future work must assess the added value from using multivariate geostatistics by comparing several methods in terms of the complexity, error prediction, data used, and so on.

## Figures and Tables

**Figure 1 sensors-21-04717-f001:**
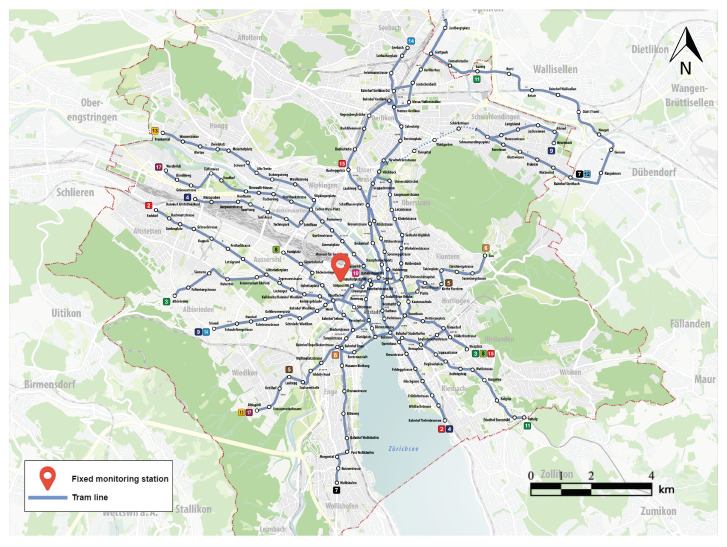
Location of the fixed station and the tram paths in the city of Zurich.

**Figure 2 sensors-21-04717-f002:**
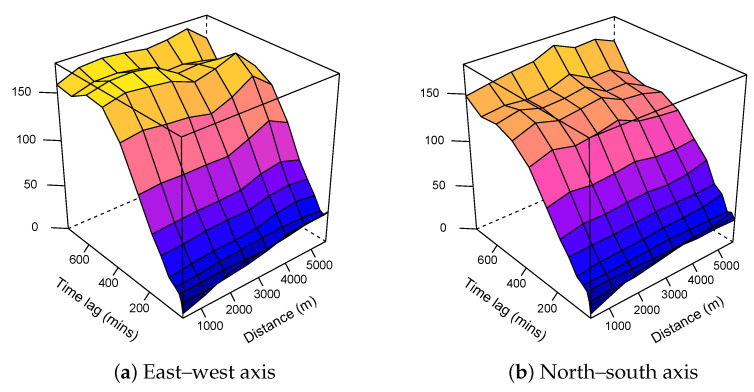
Directional spatio-temporal empirical variograms.

**Figure 3 sensors-21-04717-f003:**
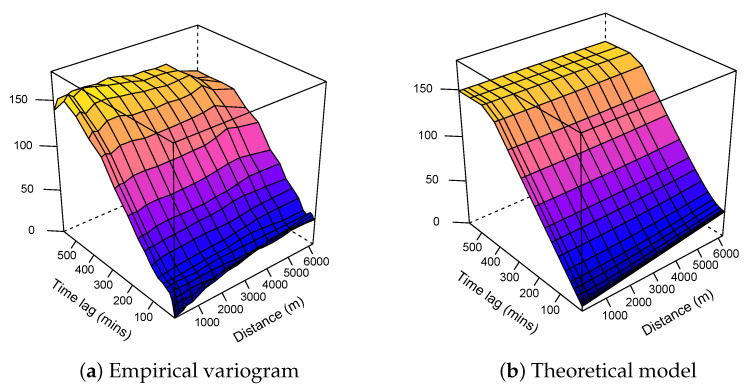
Spatio-temporal variograms associated with the ordinary kriging model.

**Figure 4 sensors-21-04717-f004:**
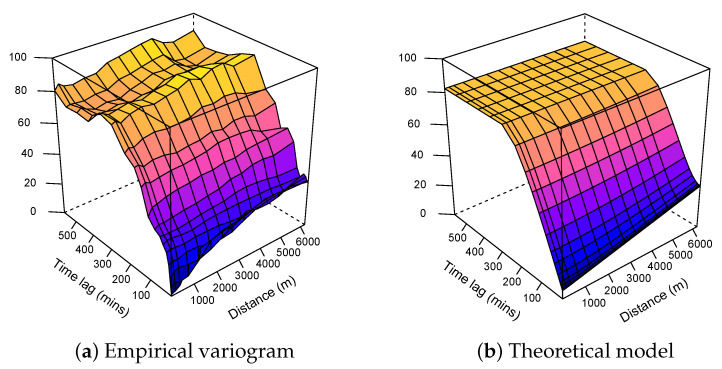
Spatio-temporal variograms associated with the simple kriging model.

**Figure 5 sensors-21-04717-f005:**
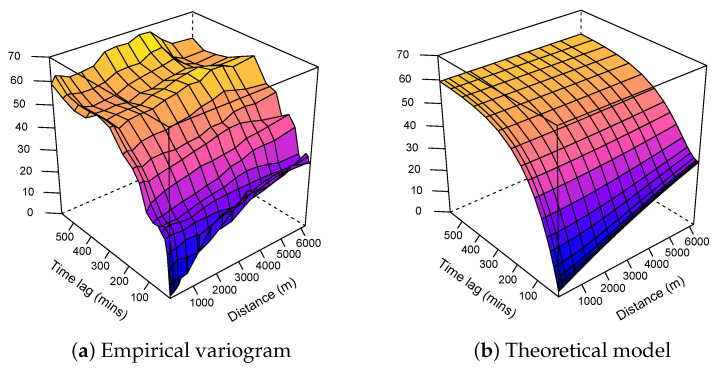
Spatio-temporal variograms associated with the kriging with external drift model.

**Figure 6 sensors-21-04717-f006:**
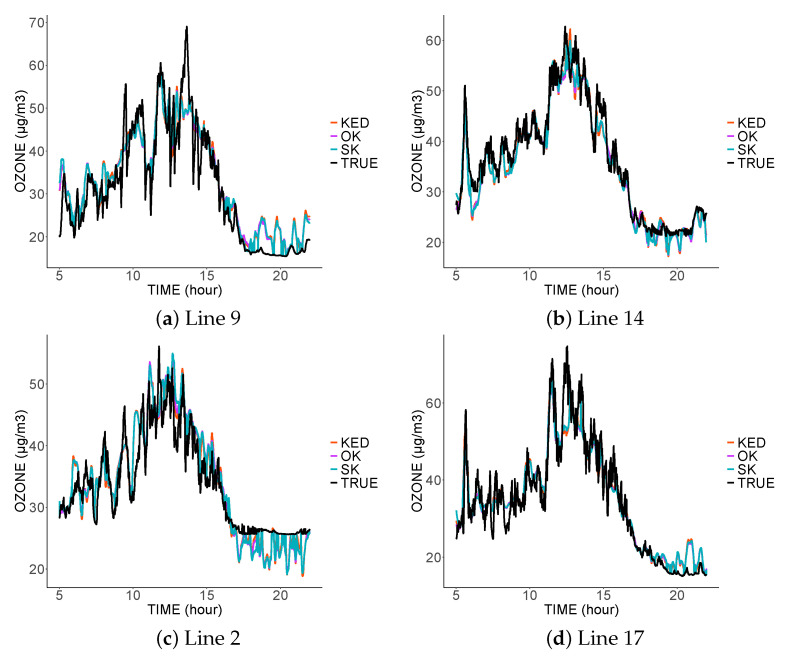
Comparison between the predictions and real values from four tram lines on 4 March 2016 in the first scenario.

**Figure 7 sensors-21-04717-f007:**
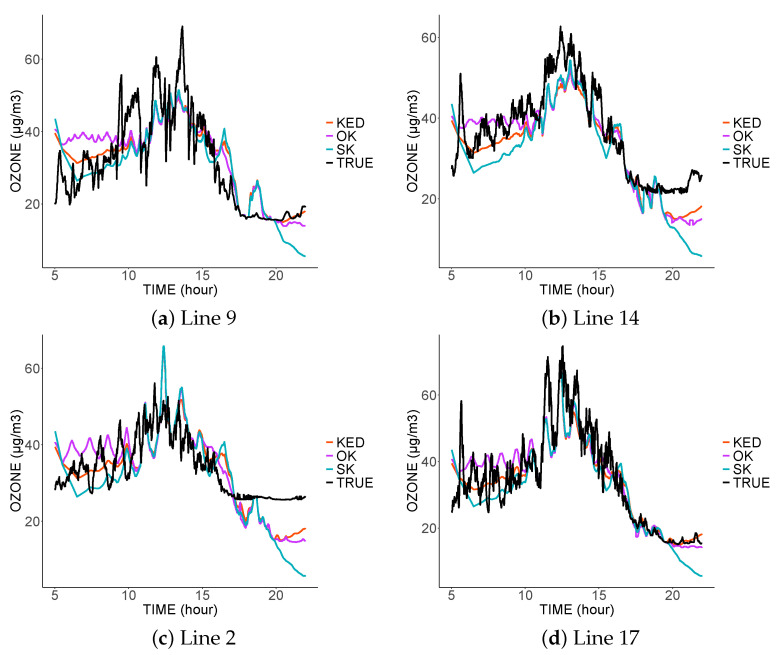
Comparison between the predictions and real values from four tram lines on 4 March 2016 in the second scenario.

**Figure 8 sensors-21-04717-f008:**
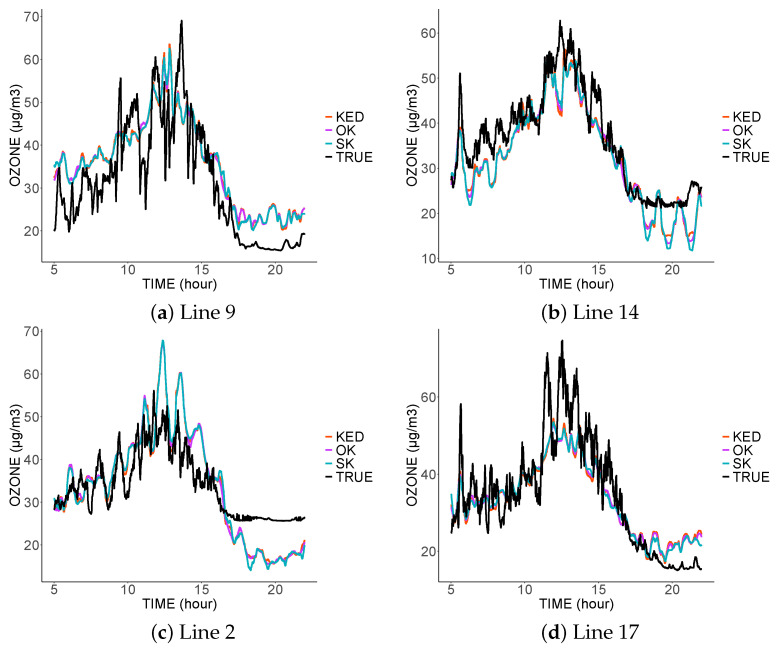
Comparison between the predictions and real values from four tram lines on 4 March 2016 in the third scenario.

**Figure 9 sensors-21-04717-f009:**
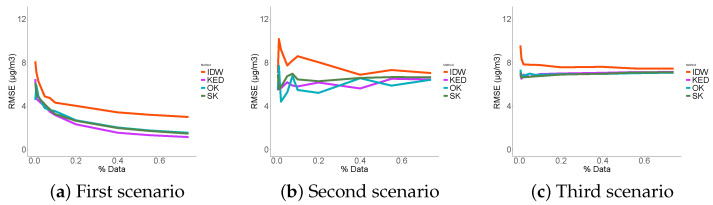
RMSE.

**Figure 10 sensors-21-04717-f010:**
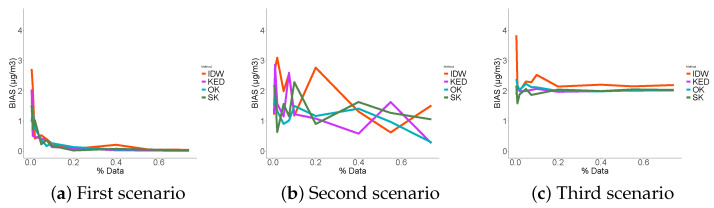
BIAIS.

**Figure 11 sensors-21-04717-f011:**
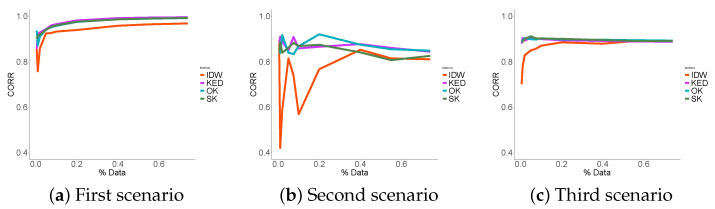
Correlations.

**Figure 12 sensors-21-04717-f012:**
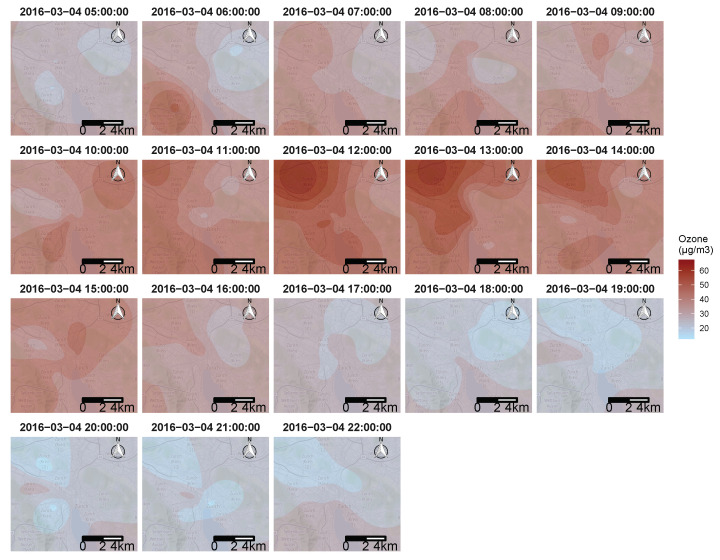
The resulting ozone concentrations maps from the KED estimator in Zurich, here shown for 4 March 2016. From 5 a.m. to 10 p.m.

**Figure 13 sensors-21-04717-f013:**
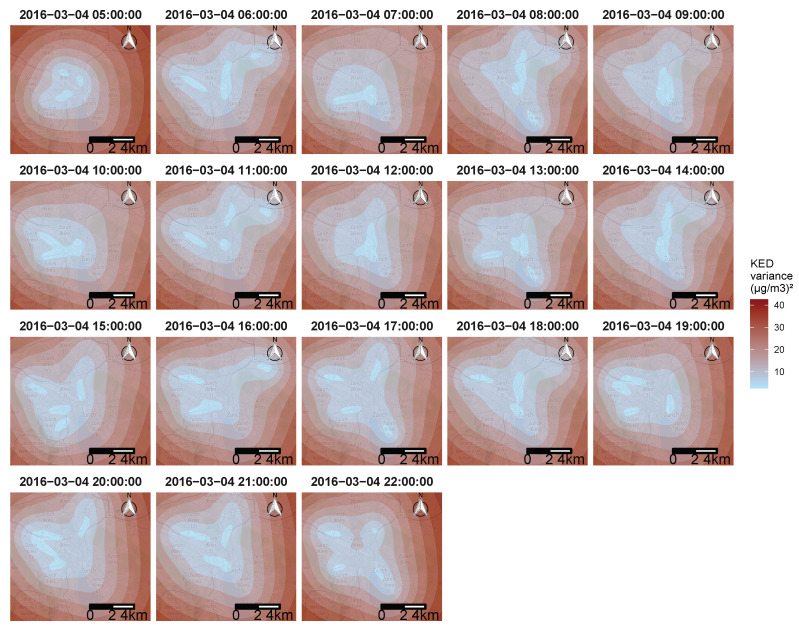
Resulting variance estimator maps for the KED estimator in Zurich, here shown for 4 March 2016. From 5 a.m. to 10 p.m.

**Table 1 sensors-21-04717-t001:** Mapping air quality studies using mobile sensors. UFP stands here for ultrafine particles, LUR for land-use regression, ANN for artificial neural network and PMx for particles smaller than x microns in diameter.

Article	Method	Area	Pollutant	Sensor Carrier
Marjovi et al. [34]	LUR, machine learning (ANN)	Lausane, Switzerland	UFP	Bus
Hart et al. [35]	LUR	Texas, USA	PM_2.5_	Bike
Apte et al. [36]	Reduction algorithm	Oakland, USA	NO, NO_2_, BC	Car
Hasenfratz et al. [37]	LUR	Zurich, Switzerland	UFP	Tram
Hasenfratz et al. [27]	LUR	Zurich, Switzerland	UFP	Tram
Marjovi et al. [38]	LUR, Probabilistic Graphical Model	Lausanne, Switzerland	UFP	Bus
Li et al. [39]	Kriging	Zurich, Switzerland	UFP	Tram
Lim et al. [40]	LUR, machine learning	Seoul, South Korea	PM_2.5_	Pedestrian
Adams et al. [41]	ANN	Hamilton, Canada	NO_2_	Van
Hankey et al. [42]	LUR	Minneapolis, USA	BC, PM_2.5_	Bike
Gressent et al. [43]	Kriging	Nantes, France	PM_10_	Car
Do et al. [44]	Autoencoders	Antwerp, Belgium	Several pollutants	Bike
Zhang et al. [45]	Machine learning	Songdo, Korea	CO_2_, PM_2.5_, PM_10_	Car
Song et al. [46]	Machine learning	Beijing, China	PM_2.5_	Car
Van et al. [47]	LUR	Ghent, Belgium	BC	Bike
Guan et al. [48]	LUR, kriging	Oakland, California	NO2	Car
Mariano et al. [49]	Decision trees	Zurich, Switzerland	UFP	Tram
Ma et al. [50]	Machine learning	China	PM_2.5_	Car

**Table 2 sensors-21-04717-t002:** Different joint models and their respective parameters.

Method	S-P Model	K	Join Model	Sill	Nugget	Range
Simple kriging	Metric	105.16	Spheric	82.30	5.00	30,415.43
Ordinary kriging	Metric	91.18	Linear	148.8	5.00	38,073.4
Kriging with external drift	Metric	83.03	Exponential	59.86	2.00	9872.405

## Data Availability

The dataset used is this study can be found at https://zenodo.org/record/3355208, accessed on 5 January 2021.

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
