# Peer review of "Mapping Urban Air Quality from Mobile Sensors Using Spatio-Temporal Geostatistics"

_sensors, 2021, doi:10.3390/s21144717_

Round 1
Reviewer 1 Report
*) it would be interesting to provide some quantitative numerical results in the abstract.
*) The Introduction is very detailed and presents a fairly satisfactory overview of the state of the art. However, I noticed that the authors did not consider atmospheric monitoring using fuzzy techniques. In this regard, I recommend inserting at least one sentence in the text that highlights this important monitoring technique by putting the following relevant bibliographical references in the bibliography:
- M. Cacciola, D. Pellicanò, G. Megali, A. Lay-Ekuakille, M. Versaci, F.C. Morabito, Aspects about air pollution prediction on urban environment, 4th IMEKO TC19 Symposium on Environmental Instrumentation and Measurements 2013: Protection Environment, Climate Changes and Pollution Control, Lecce, 3 June 2013 - 4 June 2013, pp. 15-20
- B.W. Dionova, M.N. Mohammed, S. Al-Zubaidi, E. Yusuf, Environment indoor air quality assessment using fuzzy inference systems, ICT Express, Vol. 6, Issue 3, 2020, pp. 185-14.
*) Please explain the available data in more detail.
*) Please give a more detailed explanation of the reasons that led to the choice of the three performance indicators.
*) How was (1) obtained?
*) The simple kriging estimator is (2). Why?
Reviewer 2 Report
Major comments:
- My major concern about the manuscript is what data are actually used for the analysis. line 304 suggests that the fitted values from eqn (1) are used as sensor data for the remaining analysis, but line 330 suggests raw data from mobile sensors are used. In my opinion, the latter case is what should have been done. Some clarifications are necessary.
- More details on the cross validation should be provided. The week of 2/28-3/5 is the study period, but the predictions are all made for 3/4. How is the data on 3/5 used in the analysis. For example, scenario 1 says a proportion of data in space and time are chosen for training, does this mean the data from the week of 2/28-3/5 are selected for estimating model parameters as well as prediction/interpolation, or are they selected solely for parameter estimation? I would consider using a few days of data prior to 3/4 for parameter estimation, and select training data on 3/4 for interpolation because the temporal range seems to be around 300 min based on fig3, which means the data prior to 3/4 are not helpful in predicting the data on 3/4.
- One of the advantages of geostatistical models is to provide prediction uncertainty. However, there is no mention of uncertainty quantifications(UQ) in the results. Point prediction is interesting to see, but no one should trust any prediction without any UQ.
- Some notations can be improved
- Please clarify what symbols are you using to denote the mean. lines 325 use m to denote mean, is this a constant mean over space and time? line 330 mentioned zero mean for Z(x,t), but in eqn (2), again mu is used to denote the mean. If it is zero mean, there is no need to write down mu in eqn (2).
- in line 333, sigma^2 needs to be defined. I recommend writing down the objective function mentioned in line 333.
- Please describe how K in line 385 is estimated.
Reviewer 3 Report
This paper aims to show the prediction efficiency of variogram based spatiotemporal geostatistics in the mapping process of air quality using mobile sensors without the use of external variables other than pollution data for real-time prediction purpose. Overall, this work is meaningful. However, the current version is disorganized especially the Introduction and Methods sections.
- Introduction section: the current version is so lengthy! Please shorten and be focused. Your focus of this paper is how to use the spatiotemporal geostatistics to map the urban air quality based on mobile sensors. You do not need to review so much about the fixed pollution sensors. Please focus on the most related literatures about your research topic. What are the advantages and disadvantages of the previous studies? Then propose your specific research questions and major contributions. Please rewrite this section.
- Methods section: This section is disorganized and not easy to follow. Generally, you should follow Methods and Materials (Data, Study Area, Different methods used), Results, Discussion, and Conclusions. The authors firstly introduce the methods including the data. The detailed definition of the dataset is introduced in Subsection 2.2. Please merge them together. For Section 3, what are your contributions or just repeat the existing theory? This is also Methods. Please reorganize Sections 2 & 3 and rewrite.
- It is strange for 4.0.1 and 4.0.2?
- Figures: The authors did not follow the conventions to make maps such as Figure 1 and 12. Please carefully follow the rules to make maps (north arrow, scale bars are all missed). You do not need to add them to each subfigure. You can add them to a single subfigure if they (north arrow, scale bars) are the same.
- Some minor comments:
-
- Line 4: arise de difficulty to build mathematical models (Please double check this sentence).
- Acronym: please only use the full name for the first appearance, then use the acronym after that such as IDW (Line 200). Please check the whole paper.
Round 2
Reviewer 2 Report
Thanks for taking the efforts in revising the manuscript. It is much improved however, I still have major concern over the adjusted data rather than the raw data being used for analysis. The response from the author seems to suggest that the way of adjusting the data is so that they fit the statistical model assumptions for analysis. In my opinion, this can be too much data manipulation and in fact the adjusted data are simply time series instead of spatiotemporal data.
To make sure I understand the calibration step, here I want to first reiterate the steps and see if the authors agree. (1) The adjusted data used is basically intercept + slope * fixed station data. (2) Authors allow each sensor to have different intercept and slope to obtain multiple time series, but the underlying variability for these time series all are the same, because they are linear transformations of the fixed station.
The problem of using the adjusted data is that there are no spatial variability because they are simply functions of the fixed station that only varies over time. Therefore, it is inappropriate to use spatial temporal variogram for analysis. This also explains why fig2-5 shows only temporal correlation but no spatial correlation (variogram is flat over distance) for each time lag.
I think the appropriate analysis could use the adjusted data as the overall (temporal) mean functions for the mobile sensors. Take the raw data and subtract the overall mean, then apply kriging on the residuals (to estimate local spatial variability). Then perform the different kriging methods shown in the paper. The assumption would be that each sensor follow same time trend as the fixed station (after linear transforming the fixed stn data to account for bias). Once the time variability is removed, the residuals represent local spatial variability.
Reviewer 3 Report
I do not have further questions.
Author Response
Thank you for your revision and your remarks which contributed a lot to the improvement of the paper.